# Byzantine-Tolerant Distributed Multiclass Sparse Linear Discriminant Analysis

Yajie Bao[1]        Weidong Liu[1,2]        Xiaojun Mao[*1]        Weijia Xiong[3]

[1]School of Mathematical Sciences, Shanghai Jiao Tong University, Shanghai, China
[2] MoE Key Lab of Artificial Intelligence, Shanghai Jiao Tong University, Shanghai, China
[3]School of Public Health, The University of Hong Kong, Hong Kong, China

## Abstract

Communication cost and security issues are both important in large-scale distributed machine learning. In this paper, we investigate a multiclass sparse classification problem under two distributed systems. We propose two distributed multiclass sparse discriminant analysis algorithms based on mean-aggregation and median-aggregation under the normal distributed system or Byzantine failure system. Both of them are computation and communication efficient. Several theoretical results, including estimation error bounds, and support recovery, are established. With moderate initial estimators, our iterative estimators achieve a (near-)optimal rate and exact support recovery after a constant number of rounds. Experiments on both synthetic and real datasets are provided to demonstrate the effectiveness of our proposed methods.

## 1 INTRODUCTION

Multiclass classification is one of the most important topics in machine learning and plays a crucial role in many fields, such as facial recognition, text mining, and gene analysis [Heisele et al., 2001, Zhang et al., 2015, Ramaswamy et al., 2001].

Linear discriminant analysis (LDA) is a useful tool in classification problem, which aims to find linear discriminant directions to separate the samples from different classes. We consider the random variable $\boldsymbol{X}$ and its label $Y$, where $\boldsymbol{X} \in \mathbb{R}^p$ is a multivariate normal random variable with mean $\boldsymbol{\mu}_k$ and covariance matrix $\boldsymbol{\Sigma}$ when $Y = k$ for $k = 1, 2, ..., K$. Let $\pi_k = \mathbb{P}(Y = k)$ be the prior probability that variable $\boldsymbol{X}$ is observed from Class $k$, the oracle

Bayes rule under the LDA model can be written as

$$\widehat{Y} = \arg\max_k \left\{ \boldsymbol{\theta}_k^{*\top} \left( \boldsymbol{X} - \frac{\boldsymbol{\mu}_k + \boldsymbol{\mu}_1}{2} \right) + \log \frac{\pi_k}{\pi_1} \right\},$$

where $\boldsymbol{\theta}_k^* = \boldsymbol{\Sigma}^{-1}(\boldsymbol{\mu}_k - \boldsymbol{\mu}_1)$ for $k = 1, 2, ..., K$ denotes Fisher's discriminant directions. In practice, we need to estimate $\boldsymbol{\mu}_1$, $\boldsymbol{\mu}_k$, and $\boldsymbol{\Sigma}$ to obtain the estimation of $\boldsymbol{\theta}_k^*$. Given independent samples $\{(\boldsymbol{X}_i, Y_i), i = 1, 2, ..., N\}$ from $K$ classes and denote $\sum_{i=1}^{N} \mathbb{I}(Y_i = k) = N_k$, the classical estimators of Fisher's discriminant directions are given by $\widehat{\boldsymbol{\theta}}_k = \widehat{\boldsymbol{\Sigma}}^{-1}(\widehat{\boldsymbol{\mu}}_k - \widehat{\boldsymbol{\mu}}_1)$ for $k = 2, ..., K$ where $\widehat{\boldsymbol{\mu}}_k = \sum_{\{i:Y_i=k\}} X_i / N_k$ and

$$\widehat{\boldsymbol{\Sigma}} = \frac{1}{N} \sum_{k=1}^{K} \sum_{\{i:Y_i=k\}} (\boldsymbol{X}_i - \widehat{\boldsymbol{\mu}}_k)(\boldsymbol{X}_i - \widehat{\boldsymbol{\mu}}_k)^{\top}.$$

Then the classical discriminant rule is

$$\widehat{Y} = \arg\max_k \left\{ \widehat{\boldsymbol{\theta}}_k^{\top} \left( \boldsymbol{X} - \frac{\widehat{\boldsymbol{\mu}}_k + \widehat{\boldsymbol{\mu}}_1}{2} \right) + \log \frac{\widehat{\pi}_k}{\widehat{\pi}_1} \right\},$$

where $\widehat{\pi}_k = N_k / N$. It has been shown to be both theoretical and practical efficient in the classical fixed dimensionality regime. Nevertheless, the classical linear discriminant rule performs poorly (no better than random guessing) when the dimensionality $p$ increases as the sample size $N$ [Bickel and Levina, 2004]. The main reason is that the sample covariance matrix will be ill-conditioned in such a case. Another related problem is over-fitting, and it leads to great performance loss to the model.

Due to the rapid growth in the size of datasets and resource sharing, there has been tremendous interest in developing distributed machine learning methods in recent years. However, not many works focus on sparse LDA in a distributed environment. The main challenge of distributed estimation is the communication cost. The existing sparse LDA algorithms require constructing an overall sample covariance matrix, which is unrealistic in the distributed system when $p$ is large since the bandwidth of the local machine is limited.

---

*Corresponding author.

*Accepted for the 38th Conference on Uncertainty in Artificial Intelligence* (UAI 2022).

To reduce communication cost, Tian and Gu [2017] proposed a distributed sparse LDA algorithm that only required one round of communication. To the best of our knowledge, it is also the only existing distributed sparse LDA algorithm. Despite this, the computational issue is salient for each local machine. The algorithm involves solving large-scale linear programming and estimating the inverse of the covariance matrix, which is computation expensive in the high-dimensional setting. Furthermore, the convergence rate in Tian and Gu [2017] will be sub-optimal when the number of local machines is large.

Another concerning issue in distributed machine learning is security. Most distributed machine learning algorithms require a master machine to aggregate the information from local machines, which are susceptible to errors due to unpredictable and potential attacks. The security issue is more prominent in large-scale distributed systems, such as Federated Learning [Konečnỳ et al., 2016]. Byzantine failure is used to model the local machine's inherent abnormal behavior, which means that some local machines may send wrong messages or behave completely arbitrarily [Lamport et al., 1982]. The algorithm in Tian and Gu [2017] takes simple averaging aggregation of the information from local machines, which is highly non-robust in the Byzantine failure system. We note that there are several works related to the robust LDA algorithm [Zhang and Yeung, 2010, Wen et al., 2018, Nie et al., 2019], whereas these methods only work for heavy-tailed data and are not resistant to Byzantine failure. Thus it is of great interest to develop Byzantine-robust multi-classification algorithm with a theoretical guarantee.

Although there are extensive studies for distributed estimation and optimization, few are related to sparse LDA in the high-dimensional regime. Tian and Gu [2017] proposed a communication efficient distributed sparse LDA method by constructing a debiased version of linear programming discriminant (LPD) estimator [Cai and Liu, 2011] for the binary classification task. Recently, Bian et al. [2020] proposed a distributed sparse LDA method that does not require the communications of data information among different local machines. Unfortunately, no theoretical guarantee was provided in their work. More importantly, both of them are sensitive to the abnormal behaviors of local machines in the distributed system.

## 1.1 OUR CONTRIBUTIONS

To address the challenge of increasing dimensionality and Byzantine failure, we propose a new communication efficient distributed sparse LDA algorithm for distributed multi-classification problem in two different systems, respectively:

- System I: The distributed system without Byzantine failures;
- System II: There are $\alpha$ fraction Byzantine local machines, and the remaining $1-\alpha$ fraction local machines are normal.

Under System I, we propose the distributed sparse LDA method based on mean-aggregation (`Mean-DSLDA`). As for System II, the median-aggregation is applied against the potential Byzantine failure. Thus we propose the Byzantine-tolerant `Median-DSLDA`. With these two methods, we highlight the main contributions of this paper:

1. Our proposed algorithm shares the same $O(p)$ communication cost with the state-of-the-art distributed learning algorithms [Lee et al., 2017a, Wang et al., 2017].

2. Compared with Tian and Gu [2017], our proposed algorithm requires less computation complexity in the local machine.

3. The theoretical results guarantee that our proposed algorithm attains (near-)optimal statistical convergence rate and exact support recovery after a constant number of communication rounds.

4. The experiments on synthetic and real data show that our proposed algorithm converges quickly, and `Median-DSLDA` is highly robust to Byzantine failures.

## 1.2 RELATED WORK

Sparse LDA methods in the high-dimensional regime have been broadly investigated in recent years [Witten and Tibshirani, 2011, Cai and Liu, 2011, Fan et al., 2012, Mai et al., 2012, 2019]. For binary classification case, Cai and Liu [2011] and Mai et al. [2012] assumed the Bayesian discriminant direction $\boldsymbol{\Sigma}^{-1}(\boldsymbol{\mu}_2 - \boldsymbol{\mu}_1)$ is sparse then directly estimated it by using Dantzig selector [Candes and Tao, 2007] and lasso penalty respectively. Unfortunately, these methods can not be generalized to multiclass classification cases easily. For the multiclass sparse LDA problem, penalized Fisher's discriminant [Witten and Tibshirani, 2011], sparse optimal scoring [Clemmensen et al., 2011] and multiclass sparse discriminant analysis (MSDA) proposed in Mai et al. [2019] are three popular proposals. Specifically, the MSDA method simultaneously estimates all the sparse Bayesian discriminant directions by solving a quadratic group lasso problem.

Owing to the growth of sample size and dimensionality of datasets, extensive works on high-dimensional sparse distributed machine learning algorithms are proposed. A popular method for distributed sparse estimation is divide-and-conquer debiased (DC-debiased) framework proposed by Lee et al. [2017a]. Thanks to easy implementation and low communication cost, the DC-debiased scheme has been broadly deployed in several sparse estimation problems [Lv and Lian, 2022, Tian and Gu, 2017, Battey et al., 2018,

*Accepted for the 38th Conference on Uncertainty in Artificial Intelligence* (UAI 2022).

Lian and Fan, 2018]. However, the debiased operation requires estimating the inverse of the Hessian matrix, which leads to expensive computation costs for each local machine. There exists a constraint on the number of local machines for DC-debiased estimators to achieve optimal statistical convergence rate[Zhang et al., 2013, Lee et al., 2017b, Battey et al., 2018]. Wang et al. [2017] and Jordan et al. [2019] developed another framework, namely Communication-efficient Surrogate Likelihood (CSL), which refines the estimator by multi-round communication. Each local machine in the CSL framework only needs to compute and transmit gradients in each round, and then the master machine solves a penalized sub-problem. Particularly, this kind of method has no restriction on the number of machines. Relate literature based on CSL scheme are Wang et al. [2019], Fan et al. [2021], Chen et al. [2020, 2021].

To tackle potential Byzantine failures in distributed learning, some related works are proposed by letting the master machine conduct a robust aggregation on the gradient information received from local machines [Yin et al., 2018, Xie et al., 2018, Alistarh et al., 2018, Li et al., 2019, Yin et al., 2019]. The most common robust aggregation rule is median-of-means (MOM). Yin et al. [2018] established several optimal statistical rates under mild conditions for the proposed robust algorithms based on median and trimmed mean operations. Other robust aggregation rules such as marginal trimmed mean, dimensional median, Krum are also investigated in the existing literature [Xie et al., 2018, Li et al., 2019]. Tu et al. [2021b] proposed a variance-reduced version of the median-of-means aggregation procedure motivated by the composite quantile. For distributed penalized regression problems, Tu et al. [2021a] developed a Byzantine-robust least-square Lasso method.

### 1.3 NOTATION

The following notations will be used throughout the paper. For a vector $\boldsymbol{x} \in \mathbb{R}^p$, $\|\boldsymbol{x}\|_1 = \sum_{j=1}^p |x_j|$, $\|\boldsymbol{x}\|_2 = (\sum_{j=1}^p x_j^2)^{1/2}$ and $\|\boldsymbol{x}\|_\infty = \max_j |x_j|$. For a matrix $\boldsymbol{A} = (A_{i,j}) \in \mathbb{R}^{p \times p}$, spectral norm is defined by $\|\boldsymbol{A}\|_2 = \sup_{\boldsymbol{x} \in \mathbb{R}^p} \|\boldsymbol{A}\boldsymbol{x}\|_2$, $\ell_\infty$ norm is defined by $\|\boldsymbol{A}\|_\infty = \max_i \sum_{j=1}^p |A_{i,j}|$ and $|\boldsymbol{A}|_\infty = \max_{i,j} |A_{i,j}|$. For symmetric matrix $\boldsymbol{A}$, the smallest and largest eigenvalue of $\boldsymbol{A}$ are denoted by $\lambda_{\min}(\boldsymbol{A})$ and $\lambda_{\max}(\boldsymbol{A})$ respectively. For a matrix $\boldsymbol{A} \in \mathbb{R}^{m \times n}$, $\boldsymbol{A}_{ST}$ denotes the submatrix $(a_{s_i t_j})$ for $S = \{s_1, \ldots, s_r\} \subseteq \{1, \ldots, m\}$ and $T = \{t_1, \ldots, t_q\} \subseteq \{1, \ldots, n\}$. For two sequences of positive numbers $c_n$ and $d_n$, we write $c_n \lesssim d_n$ if there exists some positive constant $c$ such that $c_n \leq cd_n$ holds for sufficiently large n; and $c_n \asymp d_n$ if $c_n \lesssim d_n$ and $d_n \lesssim c_n$. For a sequence of random variables $X_n$, $X_n = O_\mathbb{P}(d_n)$ means that for any $\varepsilon > 0$ there exists some positive constant $C$ such that $\mathbb{P}(|X_n| > Cd_n) < \varepsilon$.

## 2 MODELS AND ALGORITHMS

Recall that the Bayesian discriminant directions to be estimated are $\boldsymbol{\theta}_k^* = \boldsymbol{\Sigma}^{-1}(\boldsymbol{\mu}_k - \boldsymbol{\mu}_1)$ for $k = 2, ..., K$. According to Mai et al. [2019], the contribution to discriminant from $j$-th variable of $\boldsymbol{X}$ vanishes if and only if $\theta_{2,j}^* = \cdots = \theta_{K,j}^* = 0$, which means $\theta_{k,j}^*, k = 2, ..., K$ are grouped according to $j$. Denote the support set $S$ to be $S = \{j : \theta_{kj}^* \neq 0 \text{ for some } k = 2, ..., K - 1\}$ and $s = |S|$ is the sparsity. Given independent samples $\{(\boldsymbol{X}_i, Y_i), i = 1, 2, ..., N\}$ from $K$ classes, Mai et al. [2019] proposed the multiclass sparse discriminant analysis (MSDA) method and estimated $\boldsymbol{\theta}_k^*$ for $k = 2, ..., K$ simultaneously by solving the following group lasso problem

$$\min_{\boldsymbol{\theta}_2,...,\boldsymbol{\theta}_K} \sum_{k=2}^K \left\{ \frac{1}{2} \boldsymbol{\theta}_k^\top \widehat{\boldsymbol{\Sigma}} \boldsymbol{\theta}_k - (\widehat{\boldsymbol{\mu}}_k - \widehat{\boldsymbol{\mu}}_1)^\top \boldsymbol{\theta}_k \right\} + \lambda \sum_{j=1}^p \|\boldsymbol{\theta}_{(j)}\|_2,$$

where $\boldsymbol{\theta}_{(j)} = (\theta_{2,j}, \ldots, \theta_{K,j})^\mathrm{T}$ and $\lambda > 0$ is the tuning parameter. MSDA achieves variable selection consistency in the centralized sample case. For new observation $\boldsymbol{X}_{\text{new}}$, we classify $\boldsymbol{X}_{\text{new}}$ to Class $\widehat{Y}$ if

$$\widehat{Y} = \arg\max_k \left\{ \left( \boldsymbol{X}_{\text{new}} - \frac{\widehat{\boldsymbol{\mu}}_k + \widehat{\boldsymbol{\mu}}_1}{2} \right)^\top \widehat{\boldsymbol{\theta}}_k + \log \frac{\widehat{\pi}_k}{\widehat{\pi}_1} \right\},$$

where $\widehat{\boldsymbol{\theta}}_1 = \boldsymbol{0}$.

For the ease of presentation, we suppose that all the samples $\{(\boldsymbol{X}_i, Y_i), i = 1, 2, ..., N\}$ are stored in the master machine and $M$ local machines evenly. Denote the sample index in the $m$-th machine by $\mathcal{H}_m$ for $m = 0, 1, ..., M$ where $\mathcal{H}_0$ is the master machine, then the samples in the $m$-th machine are $\{(\boldsymbol{X}_i, Y_i) : i \in \mathcal{H}_m\}$. When $Y_i = k$, the corresponding observation $\boldsymbol{X}_i$ is sampled from multivariate normal distribution $\mathcal{N}(\boldsymbol{\mu}_k, \boldsymbol{\Sigma})$ for $k = 1, 2, ..., K$. Without loss of generality, we assume the samples from $K$ classes are evenly distributed in both the master machine and $M$ local machines. Thus the sample size of Class $k$ in each machine is $n_k$. Let $n = \sum_{k=1}^K n_k$ be the sample size of the master machine then $N = n(M + 1)$ and $N_k = n_k(M + 1)$. In System II, owing to the existence of Byzantine failure machines, we assume that the master machine can never be corrupted so that we can trust the information collected in it. For the remaining local machines, some of them may collect contaminated data or send arbitrary wrong values to the master machine. Among the $M$ local machines, the fraction of Byzantine machines is denoted by $\alpha$ and the remaining $1 - \alpha$ local machines are normal. In each machine $\mathcal{H}_m$ for $m = 0, 1, ..., M$, we compute the corresponding estimators $\widehat{\pi}_{m,k} = \sum_{i \in \mathcal{H}_m} \mathbb{I}(Y_i = k)/n$, $\widehat{\boldsymbol{\mu}}_{m,k} = \sum_{\{i \in \mathcal{H}_m : Y_i = k\}} \boldsymbol{X}_i/n_k$ and

$$\widehat{\boldsymbol{\Sigma}}_m = \frac{1}{n} \sum_{k=1}^K \sum_{\{i \in \mathcal{H}_m : Y_i = k\}} (\boldsymbol{X}_i - \widehat{\boldsymbol{\mu}}_{m,k})(\boldsymbol{X}_i - \widehat{\boldsymbol{\mu}}_{m,k})^\top.$$

*Accepted for the 38^{th} Conference on Uncertainty in Artificial Intelligence* (UAI 2022).

Given initial estimators $\widehat{\boldsymbol{\theta}}_k^{(0)}$ for $k = 2, ..., K$ and motivated by CSL framework [Jordan et al., 2019], we update the estimator in the $t$-th iteration by solving the following quadratic group lasso problem

$$\min_{\boldsymbol{\theta}_2, ..., \boldsymbol{\theta}_K} \sum_{k=2}^{K} \left\{ \frac{1}{2} \boldsymbol{\theta}_k^\top \widehat{\boldsymbol{\Sigma}}_0 \boldsymbol{\theta}_k - (\widehat{\boldsymbol{\Sigma}}_0 \widehat{\boldsymbol{\theta}}_k^{(t-1)} - \boldsymbol{b}_k^{(t-1)})^\top \boldsymbol{\theta}_k \right\}$$
$$+ \lambda \sum_{j=1}^{p} \left\| \boldsymbol{\theta}_{(j)} \right\|_2, \quad (2.1)$$

where $\boldsymbol{b}_k^{(t-1)}$ is a consistent estimator of $\boldsymbol{\Sigma} \widehat{\boldsymbol{\theta}}_k^{(t-1)} - (\boldsymbol{\mu}_k - \boldsymbol{\mu}_1)$ given $\widehat{\boldsymbol{\theta}}_k^{(t-1)}$. The optimization problem (2.1) can be efficiently solved by several well studied methods, such as group coordinate descent algorithm [Yuan and Lin, 2006] and iterative soft-thresholding algorithm [Beck and Teboulle, 2009]. Thus the keystone is to construct $\boldsymbol{b}_k^{(t-1)}$ in the master machine by using the local information under different distributed systems (System I and II) efficiently and safely.

The following are two kinds of aggregations considered in our algorithms to construct $\boldsymbol{b}_k^{(t-1)}$ and estimate $\boldsymbol{\mu}_k, \pi_k$.

**Aggregation 1** (Mean-aggregation). *The master machine estimate $\boldsymbol{\mu}_d$ and $\pi_d$ by using $\widehat{\boldsymbol{\mu}}_d = \sum_{m=0}^{M} \widehat{\boldsymbol{\mu}}_{m,d}/(M+1)$ and $\widehat{\pi}_d = \sum_{m=0}^{M} \widehat{\pi}_{m,d}/(M+1)$ respectively for $d = 1, 2, ..., K$. Denote*

$$\widehat{\boldsymbol{d}}_{m,k}^{(t-1)} = \frac{1}{n} \sum_{d=1}^{K} \sum_{\{i:i \in \mathcal{H}_m, Y_i = d\}} (\boldsymbol{X}_i - \widehat{\boldsymbol{\mu}}_d)(\boldsymbol{X}_i - \widehat{\boldsymbol{\mu}}_d)^{\mathrm{T}} \widehat{\boldsymbol{\theta}}_k^{(t-1)},$$

*and construct $\boldsymbol{b}_k^{(t-1)}$ by using*

$$\widehat{\boldsymbol{b}}_k^{(t-1)} = \widehat{\boldsymbol{d}}_k^{(t-1)} - (\widehat{\boldsymbol{\mu}}_k - \widehat{\boldsymbol{\mu}}_1),$$

*where $\widehat{\boldsymbol{d}}_k^{(t-1)} = \sum_{m=0}^{M} \widehat{\boldsymbol{d}}_{m,k}^{(t-1)}/(M+1)$.*

For vectors $\boldsymbol{x}_m \in \mathbb{R}^p$, $m = 0, 1, ..., M$, the coordinate-wise median operator is denoted by cmed. Then $\boldsymbol{z} := \mathrm{cmed}\{\boldsymbol{x}_m, m = 1, 2..., M\}$ is a $p$-vector and $z_j$ is the median of $\{x_{m,j} : m = 0, 1, ..., M\}$. Then we define the following aggregator and estimators.

**Aggregation 2** (Median-aggregation). *The master machine estimate $\boldsymbol{\mu}_d$ and $\pi_d$ by using $\widetilde{\boldsymbol{\mu}}_d = \mathrm{cmed}\{\widehat{\boldsymbol{\mu}}_{m,d} : m = 0, 1, ..., M\}$ and $\widetilde{\pi}_d = \mathrm{median}\{\widehat{\pi}_{m,d} : m = 0, 1, ..., M\}$ respectively for $d = 1, 2, ..., K$. Denote $\widetilde{\boldsymbol{d}}_{m,k}^{(t-1)} = \widehat{\boldsymbol{\Sigma}}_m \widehat{\boldsymbol{\theta}}_k^{(t-1)}$ and construct $\boldsymbol{b}_k^{(t-1)}$ by using*

$$\widetilde{\boldsymbol{b}}_k^{(t-1)} = \widetilde{\boldsymbol{d}}_k^{(t-1)} - (\widetilde{\boldsymbol{\mu}}_k - \widetilde{\boldsymbol{\mu}}_1),$$

*where $\widetilde{\boldsymbol{d}}_k^{(t-1)} = \mathrm{cmed}\{\widetilde{\boldsymbol{d}}_{m,k}^{(t-1)} : m = 0, 1, ..., M\}$.*

With the help of Aggregation 1 and 2, we propose `Mean-DSLDA` and `Median-DSLDA` under System I and System II, respectively. We start by obtaining $\widehat{\boldsymbol{\mu}}_k, \widehat{\pi}_k, \widetilde{\boldsymbol{\mu}}_k$, and $\widetilde{\pi}_k$ in the master machine through one round of communication. Given the initial estimators $\widehat{\boldsymbol{\theta}}_k^{(1)}$ satisfying some mild conditions (see Section 3), the vectors $\widehat{\boldsymbol{d}}_{m,k}^{(1)}$ or $\widetilde{\boldsymbol{d}}_{m,k}^{(1)}$ are parallelly computed on the $m$-th local machine. We only need to communicate these $p$-dimension vectors to the master machine, thus the communication cost is $O(p)$. With these constructed vectors, the master machine computes $\boldsymbol{b}_k^{(1)}$ by Aggregation 1 and 2 under System I and II respectively and obtains the updated estimators $\widehat{\boldsymbol{\theta}}_k^{(1)}$ by solving (2.1). These steps can be repeated iteratively to refine the estimators at each communication round.

---

**Algorithm 1:** Distributed Multiclass Sparse Linear Discriminant Analysis (DSLDA)

**Input:** Local data sets $\{\boldsymbol{X}_i, Y_i : i \in \mathcal{H}_m\}$ for $m = 0, 1, ..., M$, the number of iterations $T$, the initial estimators $\{\widehat{\boldsymbol{\theta}}_k^{(0)} : k = 2, ..., K\}$, the tuning parameters $\lambda_t$ for $t = 1, ..., T$.

**Output:** Final estimators $\{\widehat{\boldsymbol{\theta}}_k^{(T)} : k = 2, ..., K\}$.

**for** $m = 0, 1, ..., M$ **do**
 The $m$-th machine: Compute $\widehat{\boldsymbol{\mu}}_{m,k}, \widehat{\pi}_{m,k}$ then send them to the master machine.
**end**

**The master machine:** Compute $\widehat{\boldsymbol{\mu}}_k, \widehat{\pi}_k, \widetilde{\boldsymbol{\mu}}_k$ and $\widetilde{\pi}_k$ for $k = 1, 2, ..., K$ then broadcast $\widehat{\boldsymbol{\theta}}_k^{(0)}$ and $\widehat{\boldsymbol{\mu}}_k$ to all local machines.

**for** $t = 1, 2, ..., T$ **do**
 **for** $m = 0, 1, ..., M$ **do**
  The $m$-th machine: Compute
  
  $$\begin{cases} \widehat{\boldsymbol{d}}_{m,k}^{(t-1)}, & \text{System I} \\ \widetilde{\boldsymbol{d}}_{m,k}^{(t-1)}, & \text{System II} \end{cases},$$
  
  according to Aggregation 1 and 2 then send it to the master machine.
 **end**
 **The master machine:** Construct $\boldsymbol{b}_k^{(t-1)}$ by
 
 $$\boldsymbol{b}_k^{(t-1)} \leftarrow \begin{cases} \widehat{\boldsymbol{b}}_k^{(t-1)}, & \text{System I} \\ \widetilde{\boldsymbol{b}}_k^{(t-1)}, & \text{System II} \end{cases},$$
 
 according to Aggregation 1 and 2 and obtain $\widehat{\boldsymbol{\theta}}_k^{(t)}$ by solving (2.1). Then broadcast $\widehat{\boldsymbol{\theta}}_k^{(t)}$ for $k = 2, ..., K$ to all local machines.
**end**

---

After $T$ communication rounds, we can obtain the final estimators $\widehat{\boldsymbol{\theta}}_k^{(T)}$ for $k = 2, ..., K$. Then for new observation

*Accepted for the 38$^{th}$ Conference on Uncertainty in Artificial Intelligence* (UAI 2022).

$X_{\text{new}}$, we classify $X_{\text{new}}$ to Class

$$
\begin{cases}
\arg\max_k \left( X_{\text{new}} - \frac{\widehat{\mu}_k + \widehat{\mu}_1}{2} \right)^\top \widehat{\theta}_k^{(T)} + \log \widehat{\pi}_k, & \text{System I} \\
\arg\max_k \left( X_{\text{new}} - \frac{\widetilde{\mu}_k + \widetilde{\mu}_1}{2} \right)^\top \widetilde{\theta}_k^{(T)} + \log \widetilde{\pi}_k, & \text{System II}
\end{cases}.
$$

The details of `Mean-DSLDA` and `Median-DSLDA` are described in Algorithm 1. Our proposed algorithm and the DC-debiased algorithm in Tian and Gu [2017] both require computing and storing the local covariance matrix $\widehat{\Sigma}_m$ in each local machine. In addition to this operation, the DC-debiased algorithm also needs to estimate the sparse discriminant direction and the inverse of the covariance matrix in each local machine, leading to $O(np^2)$ extra computation complexity least. In each communication round of Algorithm 1, each local machine only needs to compute $\widehat{d}_{m,k}^{(t-1)}$ or $\widetilde{d}_{m,k}^{(t-1)}$. The local computation complexity of our method is $O(Tp^2)$, which is sufficiently reduced compared with the DC-debiased algorithm since $T$ is a constant based on our theory (see the discussion of Corollary 3.1).

## 3 THEORETICAL RESULTS

In this section, we present the theoretical results of our proposed `Mean-DSLDA` and `Median-DSLDA` including the estimation error bounds and support recovery. With slightly abusing notations, we denote

$$
\theta_{\min}^* = \min \left\{ |\theta_{k,j}^*| : |\theta_{k,j}^*| \neq 0, j \in S, 2 \leq k \leq K \right\},
$$

$$
\Delta_{\min} = \min_{1 \leq k, d \leq K, k \neq d} \sqrt{(\mu_k - \mu_d)^\top \Sigma^{-1} (\mu_k - \mu_d)},
$$

and

$$
\Delta_{\max} = \max_{1 \leq k, d \leq K, k \neq d} \sqrt{(\mu_k - \mu_d)^\top \Sigma^{-1} (\mu_k - \mu_d)}.
$$

Let $Z_j^* \in \mathbb{R}^{K-1}$ be the subgradient of $\|\theta\|_2$ evaluated at $\theta_{(j)}^* = (\theta_{2,j}^*, \ldots, \theta_{K,j}^*)^{\mathrm{T}}$ and $Z_{S,k}^* = (Z_{j,k}^* : j \in S)$. Before presenting the formal results of our proposed method, we introduce the following technical assumptions for the clarity of the theoretical guarantee.

(**C1**) There exists a positive constant $c \geq 1$ such that $c^{-1} \leq \lambda_{\min}(\Sigma) \leq \lambda_{\max}(\Sigma) \leq c$. There exist some constants $c_1 > 0$ and $c_2 < \infty$ such that $\Delta_{\min} > c_1$, $\Delta_{\max} < c_2$.

(**C2**) The sample size of each class satisfies $N_1 \asymp N_2 \asymp \cdots \asymp N_K$. The dimensionality $p$ satisfies $\log p = O(n^\nu)$ with $\nu < \frac{1}{3}$. The sparsity $s$ satisfies that $s = O(n^\beta)$ with $\beta < \frac{1}{3}$. The sample size of the master machine satisfies that $n \gtrsim N^\psi$ with $0 < \psi < 1$.

(**C3**) The initial estimators $\widehat{\theta}_k^{(0)}$ for $k = 2, \ldots, K$ have the common support set $\widehat{S}^{(0)}$ and satisfy that $\max_{2 \leq k \leq K} \|\widehat{\theta}_k^{(0)} - \theta_k^*\|_2 = O_{\mathbb{P}}(a_n)$ with $a_n = o(1)$. Moreover, assume that $\mathbb{P}(\widehat{S}^{(0)} \subseteq S) \to 1$.

(**C4**) Suppose that $\Sigma$ satisfies that $\|\Sigma_{S^c S} \Sigma_{SS}^{-1}\|_\infty < \infty$ and for some $\kappa \in (0, 1)$,

$$
\max_{j \in S^c} \left\{ \sum_{k=2}^K \left( \Sigma_{j,S} \Sigma_{SS}^{-1} Z_{S,k}^* \right)^2 \right\}^{1/2} = 1 - \kappa.
$$

(**C5**) The fraction of Byzantine local machines $\alpha < \frac{1}{2}$.

Condition (**C1**) is common in sparse LDA literatures [Shao et al., 2011, Cai and Liu, 2011, Mai et al., 2012]. Condition (**C2**) is considered when establishing the support recovery consistency results, which also appears in Mai et al. [2019]. From condition (**C3**), the dimension $p$ is allowed to be greater than the local sample size $n$. Condition (**C4**) can be easily satisfied if we choose some sparse estimators obtained by local samples as the initial estimators. Condition (**C5**) guarantees the statistical consistency of the median-aggregation against Byzantine failures, similar assumption can be found in Yin et al. [2018], Tu et al. [2021a,b].

### 3.1 ESTIMATION ERROR BOUND

**Theorem 3.1.** *Suppose that conditions (**C1**)-(**C3**) and (**C5**) hold. By choosing the tuning parameter $\lambda_t =$*

$$
\begin{cases}
C \left( \sqrt{\frac{\log p}{N}} + a_n \left( \frac{s \log p}{n} \right)^{t/2} \right), & \text{System I} \\
C \left( \sqrt{\frac{\log p}{N}} + a_n \left( \frac{s \log p}{n} \right)^{t/2} + \frac{\alpha}{\sqrt{n}} + \frac{1}{n} \right), & \text{System II}
\end{cases}
$$

*for some sufficiently large positive constant $C$, we are guaranteed that*

$$
\max_{2 \leq k \leq K} \|\widehat{\theta}_k^{(t)} - \theta_k^*\|_2 = O_{\mathbb{P}}\left( \sqrt{s}\lambda_t \right), \quad (3.1)
$$

*for $k = 2, \ldots, K$ under both System I and II.*

Theorem 3.1 provides $\ell_2$ estimation error bounds after the $t$-th iteration in Algorithm 1. The first term in (3.1) is the minimax rate of $\ell_2$ error bound for (group)lasso estimators in the centralized sample case (see Raskutti et al. [2009], Bühlmann and Van De Geer [2011], Wainwright [2019]). The second term implies that the $\ell_2$ estimation error converges geometrically to the optimal order with contraction rate $\sqrt{s \log p / n}$. Note that `Median-DSLDA` has two additional terms in the convergence rate. The term $\alpha \sqrt{s/n}$ is owing to the existence of Byzantine failure machines while $\sqrt{s}/n$ results from the median-aggregation. Therefore, Theorem 3.1 also indicates that `Mean-DSLDA` is more efficient than `Median-DSLDA` under System I. Considering that mean-aggregation is not resistant to Byzantine failures, `Median-DSLDA` is preferred under System II.

**Corollary 3.1.** *Under the same conditions and settings in Theorem 3.1, if the initial estimator satisfies that*

$$
\max_{2 \leq k \leq K} \|\widehat{\theta}_k^{(0)} - \theta_k^*\|_2 = O_{\mathbb{P}}\left( \sqrt{\frac{s \log p}{n}} \right),
$$

*Accepted for the 38th Conference on Uncertainty in Artificial Intelligence* (UAI 2022).

*and the number of iteration round $T$ satisfies*

$$T \geq \frac{\log(N/n)}{\log\left(n/\left(s^2 \log p\right)\right)}, \qquad (3.2)$$

*we are guaranteed that*

$$\max_{2 \leq k \leq K} \|\widehat{\boldsymbol{\theta}}_k^{(T)} - \boldsymbol{\theta}_k^*\|_2$$

$$= \begin{cases} O_{\mathbb{P}}\left(\sqrt{\frac{s \log p}{N}}\right), & \text{System I} \\ O_{\mathbb{P}}\left(\sqrt{\frac{s \log p}{N}} + \frac{\alpha}{\sqrt{n}} + \frac{1}{n}\right). & \text{System II} \end{cases}$$

In System I, there is no constraint on the number of local machines $M$ since we only require the sample size of the master machine satisfies $n \geq N^\psi$. For System II, we require all the local machines have $O(n)$ samples to ensure the consistency of Median-aggregation. In fact, if the number of local machines satisfies $M \lesssim \sqrt{N}$ in the System II, the $\ell_2$ error bounds of Median-DSLDA becomes $O_{\mathbb{P}}(\sqrt{s \log p/N} + \alpha\sqrt{s/n})$, which cannot be improved due to the existence of Byzantine machines. In accordance with the assumption on the sparsity $s$, the right hand side of (3.2) can be bounded by

$$\frac{\log(N/n)}{\log\left(n/\left(s^2 \log p\right)\right)} \leq \frac{1 - 1/\psi}{1 - \nu - 2\beta}.$$

It connotes that our proposed method can achieve optimal convergence rate after a constant number of communication rounds.

It is worth comparing our results with Tian and Gu [2017] for the binary classification case ($K = 2$) under System I. To achieve an optimal convergence rate, they required that the number of local machines $M$ satisfies $M \lesssim \sqrt{N/\log p}/\max(s, s')$, where $s'$ is the maximum number of nonzero elements in each column of $\boldsymbol{\Sigma}^{-1}$. However, under System I, the Mean-DSLDA algorithm has no constraint on the number of local machines and does not require the sparsity of $\boldsymbol{\Sigma}^{-1}$.

## 3.2 SUPPORT RECOVERY

Due to the group lasso property, $\widehat{\boldsymbol{\theta}}_k^{(t)}$ for $k = 2, ..., K$ have the same support set. We denote the common support set of estimator $\widehat{\boldsymbol{\theta}}_k^{(t)}$ by $\widehat{S}^{(t)}$ for $t = 1, 2, ..., T$.

**Theorem 3.2.** *Suppose that conditions* (**C**1)−(**C**5) *hold, with the same choices of the tuning parameter $\lambda_t$ in Theorem 3.1, we have $\widehat{S}^{(t)} \subseteq S$ holds with probability tending to 1. Moreover, suppose that there exists a sufficiently large constant $C > 0$ such that*

$$\theta_{\min}^* \geq C \left\|\boldsymbol{\Sigma}_{SS}^{-1}\right\|_\infty \lambda_t, \qquad (3.3)$$

*then we have $\widehat{S}^{(t)} = S$ with probability tending to 1 for both Mean-DSLDA and Median-DSLDA.*

Theorem 3.2 guarantees the exact support recovery consistency of Algorithm 1 under the beta-min condition (3.3). Note that the beta-min condition becomes weaker as iteration round $t$ increases. And if $T$ satisfies (3.2), the beta-min condition of Mean-DSLDA in Algorithm 1 under System I will reduce to $\theta_{\min}^* \geq C\|\boldsymbol{\Sigma}_{SS}^{-1}\|_\infty \sqrt{\log p/N}$, which coincides with the order in Wainwright [2009], Mai et al. [2019].

## 4 SIMULATION RESULTS

In this section, we will investigate the numerical performance of the proposed Byzantine-tolerant distributed sparse LDA method on synthetic data. Three metrics are used to evaluate the performance of algorithms: the average $\ell_2$ estimation error $\sum_{k=2}^K \|\widehat{\boldsymbol{\theta}}_k - \boldsymbol{\theta}_k^*\|_2/K$, the misclassification rate and the $F_1$ score. The $F_1$ score is defined as

$$F_1 = 2 \cdot \frac{\text{precision} \cdot \text{recall}}{\text{precision} + \text{recall}},$$

where precision $= |\widehat{S} \cap S|/|\widehat{S}|$ and recall $= |\widehat{S} \cap S|/|S|$ and $\widehat{S}$ is the support set of $\widehat{\boldsymbol{\theta}}_k$. In the following experiments, we generate a training set of size $N$, a validation set of size $1,000$ and a test set of size $1,000$ independently, then randomly partition the training set into $M + 1$ machines (including the master machine) evenly. The validation set is used to choose tuning parameter and the test set is used to compute misclassification rate. All the results are averaged over 200 independent trails.

### 4.1 MULTI-CLASS TASK

The generation of synthetic data is as follows. Denote the label of each class by $k$ for $k = 1, 2, ..., K$. We set $K = 5$, $p = 600$, $\beta_{jk} = 1.6$ for $j = 2k - 1, 2k; k = 1, ..., K$ and $\beta_{jk} = 0$ otherwise. The covariance matrix is $\boldsymbol{\Sigma} = (\sigma_{ij})_{p \times p}$ where $\sigma_{ij} = 0.5^{|i-j|}$. Let $\boldsymbol{\mu}_k = \boldsymbol{\Sigma}\boldsymbol{\beta}_k$ and $\boldsymbol{\theta}_k^* = \boldsymbol{\beta}_k - \boldsymbol{\beta}_1$, then the support set $S = \{1, 2, ..., 10\}$. In each machine, we generate Class $k$ samples independently from $\mathcal{N}(\boldsymbol{\mu}_k, \boldsymbol{\Sigma})$ for $k = 1, 2, ..., K$ with equal sample size. For Byzantine local machines, the label $Y$ is replaced by $6 - Y$. The master machine is normal in our setting. For comparison, we also run MSDA algorithm [Mai et al., 2019] using the centralized data, which is abbreviated as C-MSDA. The results are recorded by taking averages over 100 independent trials.

In the first experiment, we investigate the communication rounds of our proposed method needed to achieve numerical convergence. We set the total sample size $N$ as 50,000, and the number of machines (including the master machine) is 100. In System II, the fraction of Byzantine local machines is $\alpha = 0.2$. The trajectories of three evaluation metrics are presented in Figure 1, where the horizontal lines represent the results of C-MSDA. As we can see, all metrics of Mean-DSLDA diverge under System II, and

*Accepted for the 38$^{th}$ Conference on Uncertainty in Artificial Intelligence* (UAI 2022).

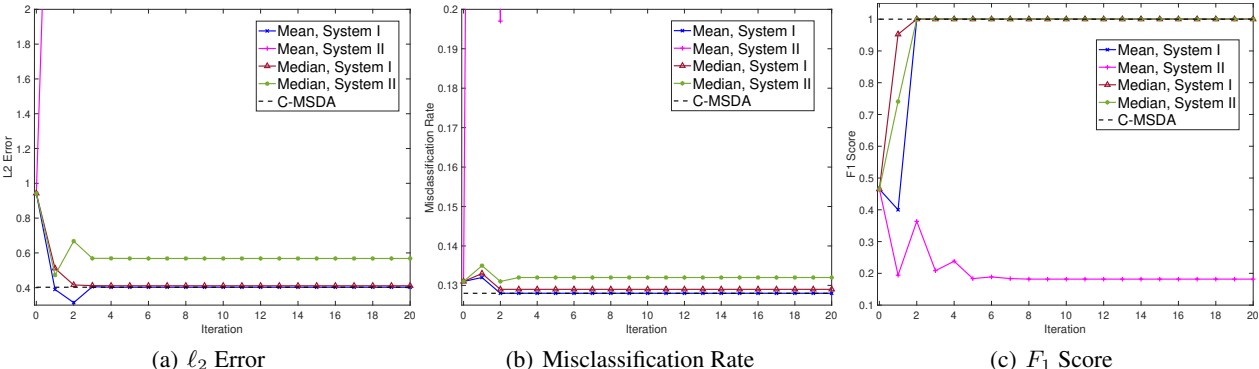

(a) $\ell_2$ Error      (b) Misclassification Rate      (c) $F_1$ Score

Figure 1: The evaluation metrics of `Mean-DSLDA`, `Median-DSLDA` and `C-MSDA` versus the number of iterations under two systems. The total sample ise $N$ is 50,000 and the number of machines is 100. The dimension is $p = 100$. The fraction of Byzantine machines in System II is $\alpha = 0.2$.

Table 1: The evaluation metrics of `Mean-DSLDA` and `Median-DSLDA` under different number of machines. The local sample size is fixed as $n = 200$ and the dimension is $p = 500$. The faction of Byzantine machines in System II is $\alpha = 0.2$.

| | L2 Error | | | | Misclassification rate | | | | F1 Score | | | |
| | System I | | System II | | System I | | System II | | System I | | System II | |
| $m$ | Mean | Median | Mean | Median | Mean | Median | Mean | Median | Mean | Median | Mean | Median |
|---|---|---|---|---|---|---|---|---|---|---|---|---|
| 100 | 0.645 | 0.677 | >10 | 0.683 | 0.135 | 0.135 | 0.615 | 0.135 | 1 | 1 | 0.278 | 1 |
| 200 | 0.698 | 0.709 | >10 | 0.748 | 0.130 | 0.131 | 0.577 | 0.130 | 1 | 1 | 0.330 | 1 |
| 300 | 0.666 | 0.696 | >10 | 0.700 | 0.129 | 0.130 | 0.619 | 0.130 | 1 | 1 | 0.275 | 1 |
| 400 | 0.663 | 0.666 | >10 | 0.677 | 0.130 | 0.135 | 0.679 | 0.135 | 1 | 1 | 0.133 | 1 |
| 500 | 0.681 | 0.694 | >10 | 0.716 | 0.128 | 0.128 | 0.706 | 0.127 | 1 | 1 | 0.155 | 1 |

`Median-DSLDA` shows robustness against Byzantine failure. In addition, the evaluation metrics of `Mean-DSLDA` (under System I) and `Median-DSLDA` converge numerically within 5 communication rounds, which corroborates the statement in Corollary 3.1. The difference of $\ell_2$ error between our proposed two methods and `C-MSDA` is tiny, which indicates the performance loss caused by distributed estimation is negligible in two systems.

In the second experiment, we investigate the effect of the number of local machines on our proposed algorithm. The local sample size is fixed as $n = 200$, and the number of machines (including the master machine) varies from 100 to 500. We summarize the averaged results in Table 1. It implies that both `Mean-DSLDA` (under System I) and `Median-DSLDA` are not sensitive to the number of machines since our proposed method can attain an optimal convergence rate without the constraint on the number of local machines.

In the third experiment, we run `Median-DSLDA` under System II with the fraction of Byzantine local machines varying from 0 to 0.2. The total sample size $N$ is fixed as 20,000, and the number of machines $M+1$ is 100. We report averages and standard deviations of three evaluation metrics

after the 5-th iteration in Table 2. There is no significant performance deterioration for `Median-DSLDA` with the increasing fraction of Byzantine local machines.

### 4.2 BINARY-CLASS TASK

For binary-class task, we compare our method with the debiased procedure in Tian and Gu [2017], which is abbreviated as `DC-LPD`. For the fairness of comparison, we follow the same data generation regime in `DC-LPD`. From the results in Table 3, it can be seen that our method has a better performance over `DC-LPD`, and the computational superiority is salient.

## 5 REAL DATA

In this section, we use the MNIST dataset[1] and ISOLET dataset[2] to verify the performance of our proposed algorithm in real data. A brief description of the two datasets is given in Table 4. We randomly divide the training sets

---

[1]http://yann.lecun.com/exdb/mnist/
[2]https://archive.ics.uci.edu/ml/datasets/isolet

*Accepted for the 38th Conference on Uncertainty in Artificial Intelligence* (UAI 2022).

Table 2: The evaluation metrics and their standard deviations (in parentheses) of `Median-DSLDA` under System II. The total sample ise $N$ is 20,000 and the number of machines (include the master machine) is 100. The dimension is $p = 500$.

|  | $\alpha = 0$ | $\alpha = 0.05$ | $\alpha = 0.1$ | $\alpha = 0.15$ | $\alpha = 0.2$ |
|---|---|---|---|---|---|
| $\ell_2$ Error | 0.687 (0.237) | 0.691 (0.243) | 0.696 (0.181) | 0.702 (0.189) | 0.717 (0.198) |
| Misclassification Rate | 0.134 (0.034) | 0.134 (0.031) | 0.134 (0.027) | 0.133 (0.026) | 0.134 (0.027) |
| $F_1$ Score | 0.995 (0.053) | 0.996 (0.054) | 0.996 (0.047) | 0.996 (0.052) | 0.996 (0.053) |

Table 3: The average evaluation metrics and local computational time. The total sample size is 10,000 and the number of machines is 20. The dimension is $p = 200$.

|  | Misclassification Rate | | | $\ell_2$ Error | | | $F_1$ Score | | | Running Time (s) | | |
|---|---|---|---|---|---|---|---|---|---|---|---|---|
| $p$ | 300 | 400 | 500 | 300 | 400 | 500 | 300 | 400 | 500 | 300 | 400 | 500 |
| `Mean-DSLDA` | 0.161 | 0.168 | 0.166 | 0.467 | 0.440 | 0.446 | 0.984 | 0.979 | 0.98 | 1.69 | 2.15 | 3.08 |
| `DC-LPD` | 0.166 | 0.171 | 0.170 | 1.190 | 1.161 | 1.239 | 0.714 | 0.733 | 0.722 | 37.79 | 92.03 | 203.84 |

of the MNIST dataset and ISOLET dataset into 20 and 10 machines, respectively (including the master machine) with an equal sample size. For Byzantine local machines, we use a similar adversarial setting in synthetic data experiments. Then we conduct Algorithm 1 by setting the iteration step $T = 20$. The tuning parameter $\lambda_t$ in each iteration is selected by five-fold cross-validation.

Table 4: Data description of MNIST and ISOLET.

| Dataset | $K$ | Training size | Test size | Dimension | Label |
|---|---|---|---|---|---|
| MNIST | 10 | 60,000 | 10,000 | 784 | 0-9 |
| ISOLET | 26 | 6,238 | 1,559 | 617 | 1-26 |

The experiment results are reported in Figure 2. As we can see, the test errors of our proposed methods decrease dramatically after the first communication round, then becomes stable in the future iterations. Under System I, the test error of `Mean-DSLDA` is lower than `Median-DSLDA`. Under System II, the classification performance of `Mean-DSLDA` is severely affected by the Byzantine machines. In addition, the utility of `Median-DSLDA` does not degrade significantly under System II.

# 6 CONCLUSIONS

In this paper, we proposed a communication efficient distributed sparse linear discriminant analysis (`Mean-DSLDA`) algorithm under a normal distributed system and its Byzantine-tolerant version (`Median-DSLDA`) for the multi-classification problem. Compared with the existing distributed sparse LDA algorithm, our proposed algorithm sufficiently reduces the computation complexity of each local machine. To achieve the optimal statistical convergence rate, `Mean-DSLDA` does not require any restrictions on the number of local machines $M$, which can be applied in a

large scale distributed system. Experiments on synthetic and real data corroborate the theoretical results and the superiority of `Median-DSLDA` against Byzantine failures.

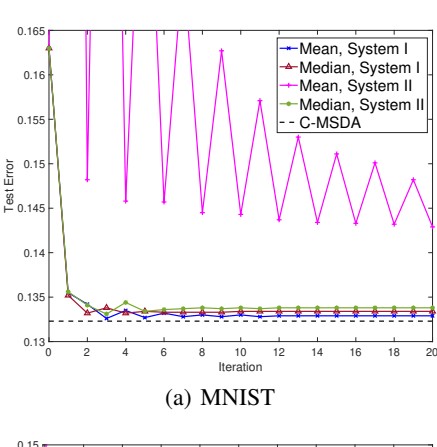

(a) MNIST

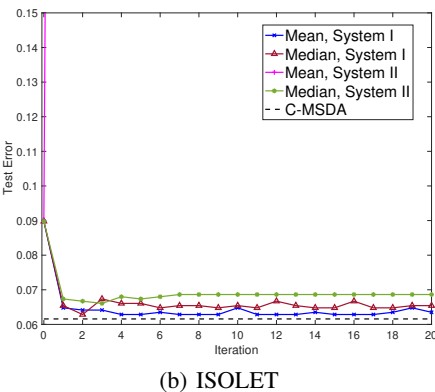

(b) ISOLET

Figure 2: The test classification error versus the number of iterations on real data. The fraction of Byzantine machines under System II is $\alpha = 0.1$. The numbers of machines in MNIST and ISOLET are respectively 100 and 10.

*Accepted for the 38$^{th}$ Conference on Uncertainty in Artificial Intelligence* (UAI 2022).

## Acknowledgements

Weidong Liu's research is supported by National Program on Key Basic Research Project (2018AAA0100704), NSFC Grant No. 11825104 and 11690013, Youth Talent Support Program, Shanghai Municipal Science and Technology Major Project (2021SHZDZX0102). Xiaojun Mao's research is supported by NSFC Grant No. 12001109 and 92046021, the Science and Technology Commission of Shanghai Municipality grant 20dz1200600.

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
