# OpenReview forum: "Byzantine-Tolerant Distributed Multiclass Sparse Linear Discriminant Analysis"
_auai.org/UAI/2022/Conference — UAI 2022 Poster_

### Official Review · Reviewer_rUV1 · 2022-03-26

**Q2(1) Originality/Novelty:** 2
**Q2(2) Significance/Impact:** 2
**Q2(3) Correctness/Technical Quality:** 2
**Q2(6) Clarity Of Writing:** 3
**Q6 Overall Score:** 6
**Q8 Confidence In Your Score:** 3

**Q1 Summary And Contributions:**

In this paper, authors propose two distributed multiclass sparse discriminant analysis algorithms based on meanaggregation and median-aggregation for distributed machine learning, along with several theoretical analysis. Experiments demonstrate the effectiveness of the proposed methods on both synthetic and real datasets.
The research topic is interesting and the corresponding theoretical analysis is given. But the comparative study doesn't show great improvement in some aspects.


**Q2 Assessment Of The Paper:**

More detailed information regarding each of these aspects is given below:

**Q2(4) Quality Of Experiments (Optional):**

2: Fair: The experimental evaluation is weak: important baselines are missing, or the results do not adequately support the main claims.

**Q2(5) Reproducibility:**

3: Good: Key resources (e.g., proofs, code, data) are available and key details (e.g., proofs, experimental setup) are sufficiently well-described for competent researchers to confidently reproduce the main results.

**Q3 Main Strengths:**

This paper provides two distributed machine learning methods;
The related theoretical analysis is clear.

**Q4 Main Weakness:**

1. the scales of the two real datasets used in this study are not really large and not convincible; and the improvement on the performance is limited.
2. the environment requirement of the algorithms are not described clearly.

**Q5 Detailed Comments To The Authors:**

Some more analysis and more data sets should be included in the experimental part.

**Q7 Justification For Your Score:**

In general, this paper is well written, but the experimental part is not good enough to maintain the analysis of the algorithms.

**Q9 Complying With Reviewing Instructions:**

1: Yes.

---

### Official Review · Reviewer_xAPe · 2022-04-10

**Q2(1) Originality/Novelty:** 2
**Q2(2) Significance/Impact:** 2
**Q2(3) Correctness/Technical Quality:** 3
**Q2(6) Clarity Of Writing:** 3
**Q6 Overall Score:** 5
**Q8 Confidence In Your Score:** 4

**Q1 Summary And Contributions:**

The authors considered the problem of linear discriminant analysis (LDA) under communication constraints and Byzantine failure. The primary contribution is proposing two algorithms for the no-failure and has-failure cases and analyzing them theoretically and empirically.

**Q2 Assessment Of The Paper:**

More detailed information regarding each of these aspects is given below:

**Q2(4) Quality Of Experiments (Optional):**

3: Good: The experimental evaluation is adequate, and the results convincingly support the main claims.

**Q2(5) Reproducibility:**

3: Good: Key resources (e.g., proofs, code, data) are available and key details (e.g., proofs, experimental setup) are sufficiently well-described for competent researchers to confidently reproduce the main results.

**Q3 Main Strengths:**

The problem considered here is fundamental and worth studying.

The proposed algorithms are more efficient than existing ones like Tian and Gu [2017]. This advantage is significant because applications like could storage often involve large data sizes, and time efficiency is thus essential.

Under certain assumptions, the theoretical guarantees are near-optimal, connoting the satisfactory performance of the algorithm.

Resilience against local machine failures and corruption is a good plus for satisfying practical needs.

**Q4 Main Weakness:**

The no-failure case, System I, is a particular case of System II; hence the authors should mention the necessity of the former. My understanding is that the no-failure scenario guarantee is only slightly better than the other due to the additional 1/n term (omitting \sqrt{s}). Unless M is as large as n, this particular term is no more significant than the leading \sqrt{\log p /N} term. So, maybe we do not need the mean-based algorithm and a separate formulation besides System II?

The assumption that the master machine is free from failure or corruption seems to simplify the problem quite a bit. I wonder if the authors have considered cases where only \alpha < 1/2 is available, and we don't know which machines, including the master one, are genuine. In the latter case, we may still borrow techniques from robust statistics to prune and filter bad samples and aggregate the information appropriately.

**Q5 Detailed Comments To The Authors:**

Please also refer to other sections. The writing is okay but not easy to follow and needs improvement. Some suggestions:

It may be better to collect all the necessary notations and present them together in a single place. Some appear at the beginning of Section 1.3, and others, like N_k and N, spread across the remaining paragraphs.

I would like to see more discussions and experimental results on the time efficiency of the algorithms. The authors claimed this attribute as the primary advantage over Tian and Gu [2017] but provided minimal discussion on this. Instead, I saw lots of comments on the algorithm's learning errors.

**Q7 Justification For Your Score:**

I think the paper is technically sound and qualifies for a meaningful contribution to distributed learning. The proposed algorithms possess optimal guarantees and satisfactory experimental performance. However, the submission has somewhat restrictive problem formulation and involves unnecessary algorithms and settings. Moreover, the authors claimed time efficiency as their primary advantage over existing works but investigated this aspect only briefly.

**Q9 Complying With Reviewing Instructions:**

1: Yes.

---

### Official Review · Reviewer_GiWC · 2022-04-30

**Q2(1) Originality/Novelty:** 3
**Q2(2) Significance/Impact:** 2
**Q2(3) Correctness/Technical Quality:** 3
**Q2(6) Clarity Of Writing:** 4
**Q6 Overall Score:** 6
**Q8 Confidence In Your Score:** 2

**Q1 Summary And Contributions:**

This paper develops a distributed sparse linear discriminant analysis (LDA) algorithm. It comes in two variants, which leverage mean and median aggregation under normal distributed system and Byzantine failure system, respectively. The paper presents theoretical results of the proposed algorithm and experiments on synthetic and real data. The two key contributions are: (1) the algorithm is more robust to Byzantine failure, and (2) it requires a lower computational complexity for local machines.

**Q2 Assessment Of The Paper:**

More detailed information regarding each of these aspects is given below:

**Q2(4) Quality Of Experiments (Optional):**

3: Good: The experimental evaluation is adequate, and the results convincingly support the main claims.

**Q2(5) Reproducibility:**

4: Excellent: Key resources (e.g., proofs, code, data) are available and key details (e.g., proof sketches, experimental setup) are comprehensively described for competent researchers to confidently and easily reproduce the main results.

**Q3 Main Strengths:**

Originality/Novelty:
- The two key contributions are, to the best of my knowledge, novel in the context of LDA and require nontrivial efforts to develop, especially for the theoretical results.

Significance/Impact:
- Byzantine failure is an important issue in distributed and federated learning, and providing a principled method to handle it in the context of LDA may be of interest to the community.

Correctness/Technical Quality:
- The paper provides theoretical results on the estimation error bounds and support recovery of the proposed algorithm, which leverage results from high dimensional statistics and appear to be sound to the best of my knowledge.

Quality Of Experiments:
- Overall, the empirical studies support the main claims.

Reproducibility:
- The proofs and code are available. The experiment details are clearly described in the paper.

Clarity Of Writing:
- The paper is well written and clear. There are quite some notations but are well introduced before using.

**Q4 Main Weakness:**

Originality/Novelty & Significance/Impact:
- The idea of performing median aggregation is not new and has been previously studied in distributed learning.
- The contribution may seem incremental, and may be considered as a combination of existing distributed LDA method and existing method to handle Byzantine failure.

Quality Of Experiments:
- The simulations can be improved by considering more parameters involved in the bound, as well as comparing both variants (mean and median) of the proposed algorithm in the simulations. More details in Q5.

**Q5 Detailed Comments To The Authors:**

- For the local machines, Section 2 mentions that the method by Tian and Gu [2017] requires $O(np^2)$ computation, and that "the local computation complexity of our method is $O(p^2)$, which is sufficiently reduced compared with Tian and Gu [2017]." However, if I understand correctly, $O(p^2)$ refers to the complexity per iteration, and therefore the total complexity of the proposed algorithm is $O(Tp^2)$. In that case, how is $O(Tp^2)$ "sufficiently reduced" relative to $O(np^2)$ by Tian and Gu [2017]?
- It might be informative to include both variants (mean and median) of the proposed algorithm together in some experiments with synthetic data (e.g., Figures 1 and 2). This could provide a sense of how the median variant performs under normal distributed system, and how the mean variant performs under failure.
- The current experiments with synthetic data (Section 4.1) only focus on different iteration number and failure rates. However, there are more other parameters involved in the theoretical bounds; thus it would be more compelling to also include simulations to verify them (e.g., varying dimensions, sample sizes).
- Can the authors provide a brief explanation (or simulations) about whether the proposed algorithm work (or how to extend it) to handle cases where the samples are not evenly distributed? This seems to be a more realistic setting in practice.
- I would suggest the authors to also explain whether the median aggregation scheme applies to the method by Tian and Gu [2017], e.g., by applying it to Eq. (3.5).

After reading rebuttal:
I appreciate the authors' detailed response. I have read the rebuttal and the other reviews, and my concerns have been well resolved.

**Q7 Justification For Your Score:**

Although the proposed algorithm might seem like an incremental extension and/or combination of existing methods (as described in Q4), I think it still appears to be a nontrivial contribution given the theoretical results.

**Q9 Complying With Reviewing Instructions:**

1: Yes.

---

### Decision · Program_Chairs · 2022-05-15

**Decision:**

Accept (Poster)

**Comment:**

Meta Review: The authors consider learning a distributed multi-class sparse LDA model, particularly including a fraction of Byzantine failures (arbitrary behavior) and providing robustness by performing median-based aggregation.  The main contribution of the work is the theoretical analysis, e.g., showing only a slight decrease in efficiency for the added robustness.

The main weakness of the paper is probably its narrow scope.  Although the general setting considered (distributed learning, fault tolerance) is of broad interest, the work is only analyzing a very simple classifier.  The novelty of the proposed estimator itself is also limited, applying a CSL framework and using median to provide robustness in the aggregation step.

Empirical assessment could also be stronger; for example, R1 provided a number of specific suggestions for more compelling evaluation.  The authors' response provided a number of additional experimental results that are helpful in understanding the impact of their method, and should be included if accepted.  However, the significance of these updates without a revision make it difficult to judge the final product.

There are also a number of minor typos in the draft that should be updated.